# Estimation of Pulmonary Arterial Pressure Using Simulated Non-Invasive Measurements and Gradient-Based Optimization Techniques

**Ryno Laubscher [1,\*], Johan Van Der Merwe [1], Philip G. Herbst [2] and Jacques Liebenberg [2]**

[1] Department of Mechanical and Mechatronic Engineering, Stellenbosch University, Private Bag X1, Matieland, Stellenbosch 7602, South Africa

[2] Division of Cardiology, Faculty of Medicine and Health Sciences, Stellenbosch University, Private Bag X1, Matieland, Stellenbosch 7602, South Africa

\* Correspondence: rlaubscher@sun.ac.za

**Abstract:** Reliable quantification of pulmonary arterial pressure is essential in the diagnostic and prognostic assessment of a range of cardiovascular pathologies, including rheumatic heart disease, yet an accurate and routinely available method for its quantification remains elusive. This work proposes an approach to infer pulmonary arterial pressure based on scientific machine learning techniques and non-invasive, clinically available measurements. A 0D multicompartment model of the cardiovascular system was optimized using several optimization algorithms subject to forward-mode automatic differentiation. Measurement data were synthesized from known parameters to represent the healthy, mitral regurgitant, aortic stenosed, and combined valvular disease situations with and without pulmonary hypertension. Eleven model parameters were selected for optimization based on 95% explained variation in mean pulmonary arterial pressure. A hybrid Adam and limited-memory Broyden–Fletcher–Goldfarb–Shanno optimizer yielded the best results with input data including valvular flow rates, heart chamber volume changes, and systematic arterial pressure. Mean absolute percentage errors ranged from 1.8% to 3.78% over the simulated test cases. The model was able to capture pressure dynamics under hypertensive conditions with pulmonary arterial systole, diastole, and mean pressure average percentage errors of 1.12%, 2.49%, and 2.14%, respectively. The low errors highlight the potential of the proposed model to determine pulmonary pressure for diseased heart valves and pulmonary hypertensive conditions.

**Keywords:** cardiovascular 0D model; pulmonary arterial pressure; gradient-based optimization; automatic differentiation

## 1. Introduction

In sub-Saharan Africa (SSA), cardiovascular diseases account for approximately 1 million deaths per year [1]. Amongst these, rheumatic heart disease (RHD), ischemic heart disease, and pulmonary arterial hypertension (PAH) are associated with high mortality rates [2]. RHD in SSA accounts for approximately 23% of worldwide deaths from this disease, where RHD typically results in aortic and mitral valve lesions that lead to valvular regurgitation and/or stenosis. Left untreated, these lesions result in cardiac decompensation through mechanisms of ventricular pressure and volume overload. In rheumatic heart disease, the presence of PAH is an independent predictor of mortality [3]; therefore, patients diagnosed with RHD and who have PAH are at higher risk compared to patients with RHD and no PAH. The accurate estimation of PAH for RHD cases is crucial for clinical diagnosis and prognostic purposes. The current gold standard for the quantification of pulmonary arterial pressure (PAP) requires the use of invasive right-heart catheterization [4], which in developing countries such as those in SSA is not readily available, remains costly, and is not without risk to the patient [5]. The clinical standard for the non-invasive estimation of

PAP utilizes transthoracic Doppler echocardiography and associated correlations, but these approaches typically yield inaccurate results [6].

In the present work, a non-invasive computational approach to estimate pulmonary arterial pressure and associated cardiovascular parameters—such as pulmonary arterial impedance, left ventricular elastance, and systemic venous impedance—is proposed. The approach utilizes non-invasive measurements, including transvalvular flow rates, systemic arterial pressures, and heart volume changes over a single heartbeat cycle, along with scientific machine learning techniques [7]. This, in turn, combines a mechanistic model of the cardiovascular system along with gradient-based optimization and forward-mode automatic differentiation.

Several researchers have recently investigated the efficacy of computational parameter estimation strategies to find unknown physiological parameters of the human cardiovascular system by using clinical measurements, 0D cardiovascular dynamic models, and optimization routines. Bjordalsbakke et al. [8] developed a 0D computer model of a human systemic loop and used non-invasive measurements and the trust-region-reflective algorithm to estimate various parameters, such as systemic compliances and left ventricular elastance. Synthetic data were generated using the 0D cardiovascular model with known parameters to gauge the accuracy of the parameter estimation workflow. The mean absolute percentage error (MAPE) between the true parameters and the estimated ones ranged between 1 and 10%. Kershavarz-Motamed [9] developed a workflow to estimate circulatory parameters using non-invasive measurements such as valvular flow rates measured using Doppler echography and systemic arterial pressures measured using an arm-cuff device. The 0D cardiovascular model was developed using MATLAB Simulink and the parameters optimized using the built-in *fmincon* function. Similarly, Huang and Ying [10] developed an online estimation algorithm used to infer the parameters of a five-component arterial 0D simulation model. The unknown parameters were estimated by minimizing the squared difference between the model predictions and the corresponding measurements, which for this work was generated synthetically using the model and the known parameters. The optimization was driven using the *fmincon* function in MATLAB. Colunga et al. [11] used actual invasive and non-invasive patient data to estimate the cardiovascular system parameters of a six-component 0D model by minimizing the differences between the model predictions and measured data using the Levenberg–Marquardt optimization routine. The workflow was capable of accurately recreating the measured pressure waveforms, but no validation was performed.

To minimize the difference between the 0D model predictions and the measurements (actual or synthetic) using gradient-based optimization methods such as Levenberg–Marquardt, a trust-region-reflective algorithm or MATLAB's *fmincon* requires the calculation of the loss function–parameter gradients. In the discussed research works, the authors applied finite differences [12,13] to calculate the required gradients. As shown in [14], the use of finite differences leads to computationally expensive and numerically unstable results due to the numerical approximation of the gradients, as discussed in [15]. Numerical instability is due to the amplification of ODE solution errors through finite difference approximations. Furthermore, for variable-timestep ODE solvers, such as the one used in the present work, finite differences can lead to incorrect derivative estimates due to the different number of timesteps used in the perturbed value evaluation $F(x + \delta)$ and actual value evaluation, $F(x)$. An alternative approach to estimate the gradients is to use automatic differentiation (AD) [15]. AD can calculate the analytical gradients using chain rules and computational graphs constructed from the mathematical operations in the computer model. The major limitation of AD is that the 0D cardiovascular model and ordinary differential equation (ODE) solver should be fully differentiable, meaning that the mathematical operations should be tracked and stored to calculate the gradients.

In the present work, a fully differentiable multicompartment cardiovascular 0D ODE computer model was developed using the *Julia 1.7.0* programming language. The proposed parameter inference model solves the specified set of equations and minimizes the

squared differences between the model predictions and non-invasive measurement data by adjusting important cardiovascular parameters. Various optimization algorithms were investigated, such as conjugate gradient descent, Adam, and limited-memory Broyden–Fletcher–Goldfarb–Shanno (L-BFGS). The loss function gradients used in the majority of these optimizers were determined using forward-mode automatic differentiation.

The purpose of the present work was to infer the PAP waveforms for healthy cases, mitral regurgitation, and aortic valve stenosis cases from synthetic, non-invasive data generated using known parameters and the 0D model. In addition, to determine the reduced set of parameters with a significant effect on the mean PAP, a local sensitivity analysis was performed. To the best of the authors' knowledge, this is the first work to directly investigate the ability of a scientific machine learning model to infer PAP values using non-invasive, clinically available measurements and a 0D cardiovascular system model accounting for the dynamics of the heart valves. To reduce the costs of deploying the proposed algorithm and to enable reproducibility, the computer models were developed using free and open-source *Julia* libraries, namely, *DifferentialEquations.jl* [16], *ForwardDiff.jl* [17] (automatic differentiation), *Optim.jl* [18] (optimization framework), and *Flux.jl* [19] (first-order gradient descent optimizers).

## 2. Materials and Methods

Figure 1 depicts the parameter inference model workflow. The model starts by initializing the unknown cardiovascular model parameters $\theta$ such as the left ventricle (LV) elastance, pulmonary arterial resistance, and systemic venous impedance; it should be noted that only the model parameters that have a significant effect on the mean PAP will be optimized, as discussed in Section 2.3. Once initialized, the important model parameters are used to simulate a single cardiac cycle using a fully differentiable 0D cardiovascular system model $H(\theta)$. Next, the model predictions $\hat{x}$ corresponding to the available non-invasive measurements $\tilde{x}$ are extracted. The extracted model predictions along with the synthetic non-invasive measurements are then fed to a loss function $L(\hat{x}, \tilde{x})$ that calculates the sum-squared difference. If the loss function is above the prescribed convergence criterion $\varepsilon$, the model then calculates the loss function gradients using forward-mode automatic differentiation and then adjusts the parameters using this information, and the process is then repeated. In the subsections below, more information relating to the 0D model, datasets, optimization parameters, and optimizers is provided.

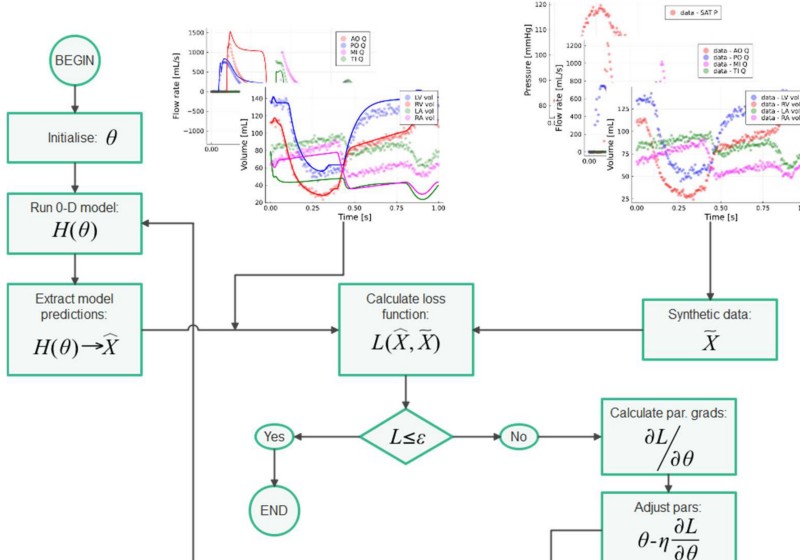

**Figure 1.** Computer model flowchart.

### 2.1. Mechanistic Model of the Cardiovascular System

Central to the pulmonary inference computer model (shown in Figure 1) is the 0D ODE model of the human cardiovascular system. In the present work, a multicompartment model including the four heart chambers—corresponding to the heart valves, pulmonary loop, and systemic loop—is developed. A layout of the cardiovascular network model is shown in Figure 2. The model is based on the work of Korakianitis and Shi [20].

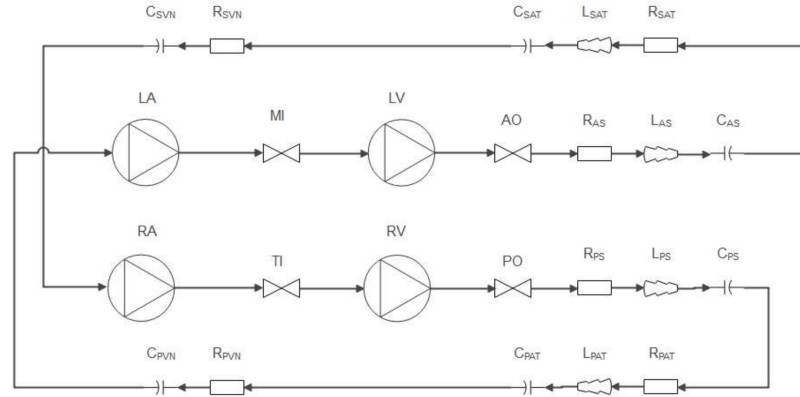

**Figure 2.** Zero-dimensional cardiovascular network model layout. Aortic—AO, mitral—MI, pulmonary—PO, tricuspid—TI, aortic sinus—AS, systemic arteries—SAT, systemic veins—SVN, pulmonary sinus—PS, pulmonary arteries—PAT, pulmonary veins—PVN.

To simulate the pressure and volume changes of the heart chambers, the mathematical model of Suga et al. [21] was applied, as shown in Equation (1), where $P_{LV}(t)$ [mmHg] is the LV pressure at time $t$, $P_{LV,0}$ is the unstressed LV pressure (set to a value of 1 for all heart chambers [22]), $e_{LV}(t)$ [s] is the LV time-varying elastance function, $V_{LV}(t)$ [mL] is the instantaneous LV volume, and $V_{LV,0}$ is the unstressed LV volume.

$$P_{LV}(t) = P_{LV,0} + e_{LV}(t)(V_{LV}(t) - V_{LV,0}) \tag{1}$$

To simulate the changes in ventricle blood volume, the mass conservation equation for an incompressible fluid can be applied to the ventricle control volume, yielding a set of ODEs for each heart chamber. The change in ventricle blood volume for the LV is shown in Equation (2), where $Q_{MI}(t)$ and $Q_{AO}(t)$ are the mitral and aortic valve volume flow rates at timestep $t$, respectively.

$$\frac{dV_{LV}}{dt} = Q_{MI}(t) - Q_{AO}(t) \tag{2}$$

The time-dependent elastance function for the LV is calculated using Equation (3), where $E_{LV,s}$ [mmHg/mL] is the LV systolic elastance and $E_{LV,d}$ is the diastolic ventricular elastance. A similar equation is used to predict the changes in right ventricle (RV) elastance.

$$e_{LV}(t) = E_{LV,d} + \frac{E_{LV,s} - E_{LV,d}}{2} f(t) \tag{3}$$

The ventricular activation function used to simulate the heart muscle contraction and relaxation was taken from the work of Bozkurt [23], and is shown in Equation (4):

$$f(t) = \begin{cases} 1 - \cos\left(\frac{t}{T_1}\pi\right) & if\ 0 \leq t < T_1 \\ 1 + \cos\left(\frac{t-T_1}{T_2-T_1}\pi\right) & if\ T_1 \leq t < T_2 \\ 0 & if\ T_2 \leq t < T \end{cases} \tag{4}$$

In Equation (4), the end time of systole is set to $T_1 = 0.3\ T$ [s] and the end time of ventricular relaxation is set to $T_2 = 0.45\ T$ [23], where $T$ is the heartbeat period, which in the

present work was fixed to a value of 1 [s]. The RV is also simulated using Equations (1)–(4), but with corresponding RV parameters (Table 1).

**Table 1.** Nominal heart model parameters (values in parentheses indicate upper and lower boundaries for sensitivity analysis and normalization) [20].

| Parameters | Left Heart | Right Heart |
|---|---|---|
| Atria | | |
| Left and right atria maximum elastances ($E_{LA,max}$, $E_{RA,max}$) | 0.25 (0.0, 1.0) | 0.25 (0.0, 1.0) |
| Left and right atria minimum elastances ($E_{LA,min}$, $E_{RA,min}$) | 0.15 (0.0, 0.5) | 0.15 (0.0, 0.5) |
| Left and right atria unstretched volumes ($V_{LA,0}$, $V_{RA,0}$) | 4.0 (1.0, 20.0) | 4.0 (1.0, 10.0) |
| Ventricles | | |
| Left and right ventricle systole elastances ($E_{LV,s}$, $E_{RV,s}$) | 2.5 (0.5, 5.0) | 1.15 (0.5, 5.0) |
| Left and right ventricle diastole elastances ($E_{LV,d}$, $E_{RV,d}$) | 0.1 (0.0, 1.0) | 0.1 (0.0, 0.5) |
| Left and right ventricle unstretched volumes ($V_{LV,0}$, $V_{RV,0}$) | 5.0 (1.0, 20.0) | 10.0 (1.0, 50.0) |

To simulate the left and right atrium (LA and RA, respectively) pressure and blood volume changes, Equations (1) and (2) are used similarly to the ventricle calculations, but with corresponding atrium parameters, as seen in Table 1. The time-dependent elastance of the left atrium is calculated as seen in Equation (5), where $E_{LA,min}$ and $E_{LA,max}$ are the minimal and maximal LA elastances, respectively. and $f_a(t)$ is the atrial contractility activation function:

$$e_{LA}(t) = E_{LA,min} + \frac{E_{LA,max} - E_{LA,min}}{2} f_a(t - D) \tag{5}$$

The atrial contractility activation function, in turn, is calculated using Equation (6), where $D = 0.04$ [s] is the time of atrial relaxation:

$$f_a(t) = \begin{cases} 0 & if\ 0 \leq t < T_a \\ 1 - cos\left(2\pi \frac{t - T_a}{T - T_a}\right) & if\ T_a \leq t < T \end{cases} \tag{6}$$

where $T_a = 0.8\ T$ is the time at the onset of atrial contraction. Table 1 presents the cardiovascular parameters used in the heart chamber models.

Typically, heart valves are modelled as simple diodes in 0D cardiovascular system models. For diode models, the valve-opening and -closure processes are assumed to be instantaneous; therefore, the inertia of the valve cusps (leaflets) is ignored. Ignoring the valve leaflet motion for diseased heart valves can lead to the prediction of higher right ventricle and pulmonary arterial pressures, as shown by [22]. Therefore, to accurately capture the pressure drop and, thus, the fluid flow through the valve, the leaflet motion should be included in the system model. The valve model implemented in the present work stems from the paper by Korakianatis and Shi [22], which includes the simulation of the valve leaflet motion by solving an angular momentum equation for each heart valve. The heart valve parameters used in the present work were tuned by the previously mentioned authors to replicate actual pressure and flow waveforms in an adult human.

To estimate the blood-flow rate through each of the four heart valves, pressure gradients across the heart valve and the valve opening area are used, as shown in Equation (7), where $i = AO, PO, TI, MI$. In Equation (7), $CQ$ is the valvular flow coefficient, which is set to $400 \left[\frac{mL}{s\ mmHg^{0.5}}\right]$ for the atrioventricular valves and $350 \left[\frac{mL}{s\ mmHg^{0.5}}\right]$ for the semilunar valves.

$$Q_i = CQ \cdot A_r(t) \sqrt{\Delta P(t)} \tag{7}$$

In the previous equation, the pressure gradient across the valve $\Delta P(t)$ is calculated using Equation (8), where $P_{in}(t)$ is the valve's upstream static pressure and $P_{ex}(t)$ is the valve's downstream pressure. For example, the aortic valve inlet pressure would be the LV

pressure $P_{LV}(t)$ and the exit pressure would be the aortic sinus pressure $P_{AS}(t)$, as shown in Figure 2.

$$\Delta P(t) = \begin{cases} P_{in}(t) - P_{ex}(t) & if\,P_{in} \geq P_{ex} \\ P_{ex}(t) - P_{in}(t) & if\,P_{ex} > P_{in} \end{cases} \tag{8}$$

In Equation (7), $A_r(t)$ is the area opening ratio of the heart valve and is defined as the fraction of flow area at a given timestep divided by the area of the valve when fully open. For the present work, the valve's opening fraction is calculated as a function of the valve's opening angle $\beta_v$, as shown in Equation (9), where $\beta_{v,max}$ is the maximum opening angle of the valve cusps:

$$A_r(t) = \frac{(1 - cos[\beta_v(t)])^2}{(1 - cos[\beta_{v,max}])^2} \tag{9}$$

To estimate the time-dependent valve-opening angle for each heart valve, the angular momentum equation is solved. To ensure that the selected ODE integrator can solve the dynamics of the valve cusp, the second-order angular momentum equation for the valve dynamics is expressed as two ODEs, as shown in Equation (10):

$$\frac{d\varphi_v}{dt} = [P_{in}(t) - P_{ex}(t)] \cdot K_p cos(\beta_v)$$
$$\frac{d\beta_v}{dt} = \varphi_v \tag{10}$$

The valvular force coefficient $K_p \left[\frac{\text{mmHg·s}^2}{\text{mL}}\right]$ was set to a constant value of 5500 for all valves, as recommended by [20]. In Equation (10), $\varphi_v$ is the valve cusp's angular velocity.

The systemic and pulmonary vasculatures are modelled using the electrohydraulic analogue equations for fluid flow in a 0D network. Each loop is modelled using 5 components that consist of inductive, capacitive, and resistive components, as shown in Figure 2. For the sake of brevity, only the systemic loop equations are provided; for more detail on the model equations, please see [24]. The flow rate through the aortic sinus and associated sinus inlet pressure are calculated using the following ODEs:

$$L_{AS} \frac{dQ_{AS}}{dt} = (P_{AS} - P_{SAT}) - R_{AS}Q_{AS} \tag{11}$$

$$C_{AS} \frac{dP_{AS}}{dt} = Q_{AO} - Q_{AS} \tag{12}$$

where $L_{AS} \left[\frac{\text{mmHg·s}^2}{\text{mL}}\right]$ is the blood flow inertia through the sinus, $Q_{AS}$ is the volume flow rate of blood through the sinus, $P_{AS}$ is the inlet sinus static pressure, $P_{SAT}$ is the arterial inlet pressure, $R_{AS} \left[\frac{\text{mmHg·s}}{\text{mL}}\right]$ is the sinus flow resistance, and $C_{AS} \left[\frac{\text{mL}}{\text{mmHg}}\right]$ is the sinus compliance. The arterial pressure and volume flow rate are simulated using Equations (13) and (14):

$$L_{SAT} \frac{dQ_{SAT}}{dt} = (P_{SAT} - P_{SVN}) - R_{SAT}Q_{SAT} \tag{13}$$

$$C_{SAT} \frac{dP_{SAT}}{dt} = Q_{AS} - Q_{SAT} \tag{14}$$

The inlet venous pressure is calculated using Equation (15), and the venous flow rate using Equation (16):

$$C_{SVN} \frac{dP_{SVN}}{dt} = Q_{SAT} - Q_{SVN} \tag{15}$$

$$Q_{SVN}R_{SVN} = P_{SVN} - P_{RA} \tag{16}$$

The vasculature parameters such as resistance and capacitance for the systemic and pulmonary loops can be found in Table 2.

**Table 2.** Systemic and pulmonary loop parameters (values in parentheses are used for the upper and lower boundaries) [20].

| Compartment | Resistance $(R)$ $\left[\frac{mmHg \cdot s}{mL}\right]$ | Inductance $(L)$ $\left[\frac{mmHg \cdot s^2}{mL}\right]$ | Capacitance $(C)$ $\left[\frac{mL}{mmHg}\right]$ |
|---|---|---|---|
| | Systemic loop | | |
| Aortic sinus (AS) | 0.003 (0.0003, 0.03) | 0.000062 $(1 \times 10^{-4}, 1 \times 10^{-3})$ | 0.08 (0.008, 0.8) |
| Systemic artery (SAT) | 0.05 (0.005, 1.0) | 0.0017 $(1.7 \times 10^{-3}, 0.017)$ | 1.6 (0.16, 3.2) |
| Systemic vein (SVN) | 0.075 (0.0075, 0.75) | 0 | 20.5 (5.0, 50.0) |
| | Pulmonary loop | | |
| Pulmonary sinus (PS) | 0.002 $(2 \times 10^{-3}, 2 \times 10^{-2})$ | 0.000052 $(1 \times 10^{-4}, 1 \times 10^{-3})$ | 0.18 (0.018, 2.0) |
| Pulmonary artery (PAT) | 0.05 (0.001, 0.1) | 0.0017 $(1.7 \times 10^{-3}, 0.017)$ | 3.8 (0.38, 6.0) |
| Pulmonary vein (PVN) | 0.006 $(6 \times 10^{-4}, 0.01)$ | 0 | 20.5 (5.0, 50.0) |

To solve the abovementioned ODEs of the cardiovascular system, the explicit Runge–Kutta solver with the Bogacki–Shampine 3/2 method was applied. The solver's relative and absolute tolerances were set to $1 \times 10^{-4}$ and $1 \times 10^{-6}$, respectively, and the maximum allowable iterations per timestep were set to $1 \times 10^6$. To numerically integrate the ODEs, certain physical constraints must be enforced on the dynamic valve model. To incorporate the discontinuities that result from the valve motion limits (fully open or closed), the following conditions were included in the simulation procedure for each valve.

$$\beta_v = \begin{cases} \beta_v = \beta_{v,max}, \ \frac{d\beta_v}{dt} = 0 \ if \ \beta_v \geq \beta_{v,max} \\ \beta_v = \beta_{v,min}, \ \frac{d\beta_v}{dt} = 0 \ if \ \beta_v \leq \beta_{v,min} \\ \beta_v \ if \ \beta_{v,min} < \beta_v < \beta_{v,max} \end{cases} \tag{17}$$

*2.2. Data and Measurements*

In the present work, synthetic data were generated using the 0D cardiovascular model and used as pseudo-clinical measurements. The benefit of this approach is that the true underlying parameters being optimized are known, and the obvious disadvantage is that one assumes that the model can capture the dynamics of an actual cardiovascular system. Nonetheless, other published authors have also followed this approach [8].

In the present work, two datasets were used as synthetic measurements. The first dataset (D1) contained the transvalvular flow rates $Q_{AO}$, $Q_{PO}$, $Q_{TI}$, and $Q_{MI}$ and the systemic arterial pressure $P_{SAT}$ for a single cardiac cycle. The second dataset (D2), in addition to the transvalvular flow rates and systemic arterial pressure, contains the heart chamber volume changes, $V_{LV}$, $V_{RV}$, $V_{LA}$, and $V_{RA}$. The motivation for using two datasets was to investigate the effects of additional non-invasive data on the model parameters' inference accuracy, as discussed in Section 3.2.

For each dataset generation run, the ODE integrator solves for multitudes of timesteps dictated by the numerical integrator accuracy control, but to replicate the actual use of the model, only $N = 200$ samples are stored and used during the parameter optimization phase. Additionally, an arbitrary amount of noise is added to the pseudo-measurement results. The standard deviation used for the normally distributed noise generation of the chamber volumes, flow rates, and arterial pressure was set as follows:

$$\sigma_{V_{lv}} = 3.2 \text{ mL}, \ \sigma_{V_{la}} = 2.5 \text{ mL}, \ \sigma_{V_{rv}} = 2.9 \text{ mL}, \ \sigma_{V_{ra}} = 2.2 \text{ mL}, \ \sigma_{P_{SAT}} = 1.1 \text{ mmHg}$$

$$\sigma_{Q_{AO}} = 2.7 \text{ mL/s}, \ \sigma_{Q_{PO}} = 2.6 \text{ mL/s}, \ \sigma_{Q_{MI}} = 2.5 \text{ mL/s}, \ \sigma_{Q_{TI}} = 2.54 \text{ mL/s}$$

Figure 3 shows the data generated for the four non-hypertensive cases analyzed in the present work, namely, the healthy case, aortic stenosis (AS) case, mitral regurgitation (MR) case, and combined AR and MR case. It should be noted that only the ventricular volume changes, valvular flow rates, and systemic arterial pressure are used as pseudo-measurements during pulmonary pressure inference, and the ventricular pressures and pulmonary pressures are merely shown for the sake of completeness.

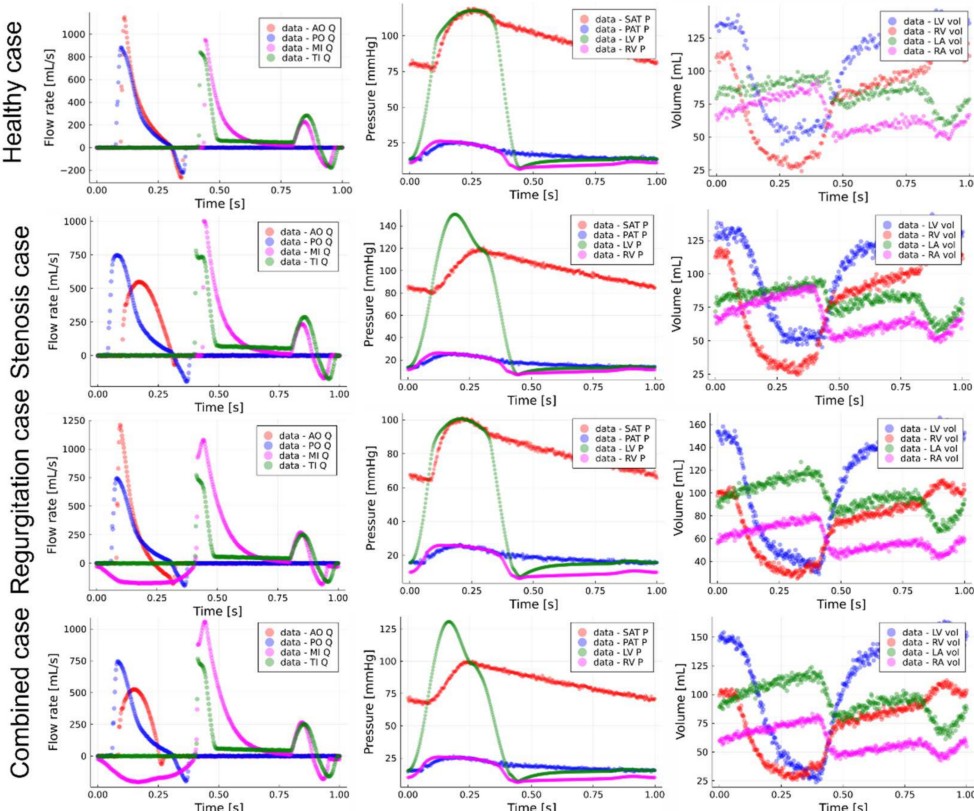

**Figure 3.** Generated data for different cases. vol—volume; SAT P—systemic arterial pressure; PAT P—pulmonary arterial pressure; LV P—left ventricular pressure; RV P—right ventricular pressure, AO Q—aortic valve flow rate, PO Q—pulmonary valve flow rate, MI Q—mitral valve flow rate, TI Q—tricuspid valve flow rate.

To clinically measure the data shown in Figure 3, different equipment can be utilized. For the present work, the following clinical measurements are proposed for further retrospective clinical studies: The brachial arterial pressure can be measured continuously using a CNAP monitor and volume clamp method, as discussed in [25]. The transvalvular flow rates should be measured using Doppler echocardiography, and the heart chamber volumes should be measured using either 3D magnetic resonance imaging (MRI) or Doppler echocardiography.

To simulate the 0D cardiovascular model and solve for the model-dependent variables such as systemic arterial pressure (Equation (14)) and aortic sinus flow rate (Equation (11)) requires the initial conditions to be known. The initial conditions vector is shown in Equation (18):

$$\hat{x}_{init} = \begin{bmatrix} V_{LV}^{init}, V_{LA}^{init}, P_{AS}^{init}, Q_{AS}^{init}, P_{SAT}^{init}, Q_{SAT}^{init}, P_{SVN}^{init}, V_{RV}^{init}, \\ V_{RA}^{init}, P_{PS}^{init}, Q_{PS}^{init}, P_{PAT}^{init}, Q_{PAT}^{init}, P_{PVN}^{init}, \beta_{AO}^{init}, \beta_{MI}^{init}, \beta_{PO}^{init}, \beta_{TI}^{init} \end{bmatrix} \tag{18}$$

In a clinical application of the proposed parameter inference model, these initial conditions should be extracted from the available non-invasive measurements. The initial transvalvular flow rates and heart chamber volumes can be directly taken as the initial entries in D1 and D2 for the respective data streams. Similarly, the initial cycle's systemic arterial pressure can be extracted from D1 and D2, and in the present work it is assumed that the initial aortic sinus pressure is equal to the initial systemic arterial pressure. The initial systemic and pulmonary arterial flow rates are approximated using Equations (19) and (20), respectively, where $SV_{LV}$ and $SV_{RV}$ are the left and right ventricular stroke volumes, respectively, which can be non-invasively estimated using Doppler echography.

$$Q_{SAT}^{init} = \frac{SV_{LV}}{T} \tag{19}$$

$$Q_{PAT}^{init} = \frac{SV_{RV}}{T} \tag{20}$$

The remaining initial conditions—namely, $P_{SVN}^{init}$, $P_{PVN}^{init}$, $P_{PS}^{init}$, and $P_{PAT}^{init}$—are difficult to accurately measure non-invasively; therefore, these parameters are optimized in conjunction with selected important model parameters (Tables 1 and 2) that significantly affect mean pulmonary arterial pressure, as discussed in Section 2.3. Since there is no substantial pressure drop between the pulmonary sinus and the pulmonary artery, the initial pulmonary arterial pressure and pulmonary sinus pressures were assumed to be equal, i.e., $P_{PVN}^{init} = P_{PS}^{init}$.

### 2.3. Local Sensitivity Analysis

A local sensitivity analysis was performed using the 0D cardiovascular model to identify model parameters (Tables 1 and 2), with a significant effect on the mean PAP. These identified parameters were then used in the optimization phase of the present work to infer the PAP waveform and estimate the true cardiovascular parameters.

To find these important parameters, the sensitivity percentages of each parameter, (designated $SP_{PAP,i}$ for the $d$ parameter) were calculated as shown in Equation (21). The top parameters making up 95% of the variance in mean PAP were then selected as the important parameters to be optimized.

$$SP_{PAP,i} = 100\% \cdot \frac{SI_{PAP,i}}{\sum_{j=1}^{n_{tot}} SI_{PAP,j}} \tag{21}$$

where $SI_{PAP,i}$ is the $i$th parameter's sensitivity index, which is calculated using Equation (22). To estimate the required gradients of the mean PAP, forward-mode AD was utilized. Forward-mode AD is capable of traversing any native *Julia* code and, therefore, is able to differentiate through the ODE integrator solution to calculate the required gradients in a computationally efficient manner [17]. The gradients were calculated around the nominal values shown in Tables 1 and 2, but seeing as the model parameters varied in orders of magnitude and units, each mean PAP gradient was multiplied by the difference between the upper $\theta_{i,ub}$ and lower $\theta_{i,lb}$ parameter boundaries to normalize the calculated parameter gradients.

$$SI_{PAP,i} = \left| \frac{\partial \left( \frac{1}{N_t^*} \sum_{j=1}^{N_t^*} P_{PAP}^j \right)}{\partial \theta_i} \right| \cdot (\theta_{i,ub} - \theta_{i,lb}) \tag{22}$$

### 2.4. Parameter Optimization

To estimate $\theta$, which minimizes the difference between the 0D model's predictions and the synthetic (pseudo)-measurements in the present work, the sum-squared error (SSE) loss function was minimized using selected optimizers. The SSE for the $j$th measurement (e.g., LV volume, systemic arterial pressure, or mitral valve flow rate) was calculated using Equation (23):

$$J\left(\hat{x}_j, \tilde{x}_j\right) = \left( \sum_{i=1}^{N} \left( \hat{x}_j^i(\overline{\theta}) - \tilde{x}_j^i \right)^2 \right)_j \tag{23}$$

where $\hat{x}_j^i$ is the $j$th simulation output at timestep $i$, and $\tilde{x}_j^i$ is the $j$th synthetically measured value (e.g., arterial pressure or LV volume) at timestep $i$. Furthermore, $\overline{\theta}$ is the parameter vector containing all of the selected important parameters, $\hat{x}_j$ is the vector of model predictions for measurement $j$, and $\tilde{x}_j$ is the vector of synthetic measurements for measurement $j$. The loss function minimized by the computer model is then simply the summation of the different measurement losses $J\left(\hat{x}_j, \tilde{x}_j\right)$, as shoen in Equation (24), where $d$ is the number of measurement streams (5 and 9 for D1 and D2, respectively, as mentioned in Section 2.2).

$$L\left(\hat{X}(\overline{\theta}), \tilde{X}\right) = \sum_{i=1}^{d} (J(\hat{x}_i, \tilde{x}_i)) \tag{24}$$

To speed up optimization convergence, the parameter and measurement datasets were normalized using min–max scaling. For the parameter vector, the upper and lower boundaries listed in Tables 1 and 2 were used. The scaling transformation of the optimization parameters can be seen in Equation (25), where $\bar{\theta}^*$ is the scaled parameter vector, $\bar{\theta}_{lb}$ is a vector of the lower boundary parameter values, and $\bar{\theta}_{ub}$ is a vector of the upper boundary parameter values.

$$\bar{\theta}^* = \frac{\bar{\theta} - \bar{\theta}_{lb}}{\bar{\theta}_{ub} - \bar{\theta}_{lb}} \tag{25}$$

The measurements and model predictions were scaled using the maximum and minimum measured values, e.g., for parameter $i$, $\max(\widetilde{x}_i)$ and $\min(\widetilde{x}_i)$. For example, Equation (26) shows the scaling of the systemic arterial pressure input waveform $P_{SAT}$ for timestep $i$:

$$\hat{x}^*_{P_{SAT},i} = \frac{\hat{x}_{P_{SAT},i} - \min(\widetilde{x}_{P_{SAT}})}{\max(\widetilde{x}_{P_{SAT}}) - \min(\widetilde{x}_{P_{SAT}})} \tag{26}$$

In the present work, three optimization strategies were employed. The first used the adaptive moment estimation (Adam) first-order optimizer. The Adam algorithm is shown in Equation (27):

$$\begin{aligned}
\bar{m} &\leftarrow \beta_1\bar{m} + (1 - \beta_1)\nabla_\theta L(\bar{\theta}) \\
\bar{s} &\leftarrow \beta_1\bar{s} + (1 - \beta_2)\nabla_\theta C(\bar{\theta}) \otimes \nabla_\theta L(\bar{\theta}) \\
\bar{m} &\leftarrow \frac{\bar{m}}{1 - \beta_1^t} \\
\bar{s} &\leftarrow \frac{\bar{s}}{1 - \beta_2^t} \\
\bar{\theta}^{new} &\leftarrow \bar{\theta} - \eta\bar{m} \otimes \sqrt{(\bar{s} + \epsilon)^{-1}}
\end{aligned} \tag{27}$$

The scaling $\bar{s}$ and momentum $\bar{m}$ matrices are initialized to 0 at the start of the Adam training algorithm, $t$ is the iteration counter, $\epsilon = 1 \times 10^{-8}$ is the smoothing term, $\beta_1$ is the momentum decay hyperparameter (and is set to 0.9), and $\beta_2$ is the scaling hyperparameter and is set to 0.999 [19]. In Equation (26), $\nabla_\theta L(\bar{\theta})$ represents the gradients of the cost function with respect to the optimization parameters. For the optimization runs, the learning rate parameter $\eta$ is fixed to a value of 0.005.

The second strategy uses the conjugate gradient descent [26] optimizer to minimize the loss function. The conjugate gradient optimizer update algorithm for the parameters is shown in Equation (28):

$$\begin{aligned}
\bar{\theta}^{new} &\leftarrow \bar{\theta} - \eta\bar{d}^{new} \\
\bar{d}^{new} &\leftarrow \nabla_\theta L(\bar{\theta}) - \gamma^{new}\bar{d}^{old}
\end{aligned} \tag{28}$$

For the first iteration, $\bar{d} = \nabla_\theta L(\bar{\theta})$. In the present work, the learning rate parameter $\eta$ was estimated per iteration using the line search proposed by Hager and Zhang [27]. The scalar variable $\gamma^{new}$ was also calculated using the method proposed by Hager and Zhang.

The third strategy applies a combination of Adam and L-BFGS [28] optimizers to minimize the loss function. For this strategy, the first 50 iterations of the optimization phase were completed using Adam, after which the model switched over to the L-BFGS. The interested reader can see [26] for more information about the L-BFGS optimization algorithm.

## 3. Results

To find the best optimization strategy and to demonstrate the ability of the proposed method to infer the PAP waveforms for diseased heart valve cases, two investigations were performed: The first looked at the effects of selected optimizers and dataset contents on the parameter estimation accuracy and pulmonary pressures for healthy heart valves. Using the best-performing optimizer of this study, we then investigated the ability of the model to

capture the PAP waveforms for synthetic data generated with induced mitral regurgitation and aortic stenosis, with and without increased pulmonary arterial impedance.

### 3.1. Local Sensitivity Analysis

Before the two above sets of results are discussed, the important model parameters selected using the local sensitivity analysis must be provided. Figure 4 shows a histogram of all of the model parameters and their respective sensitivity percentages, as calculated using Equation (21). The legend for Figure 4 is shown in Table 3.

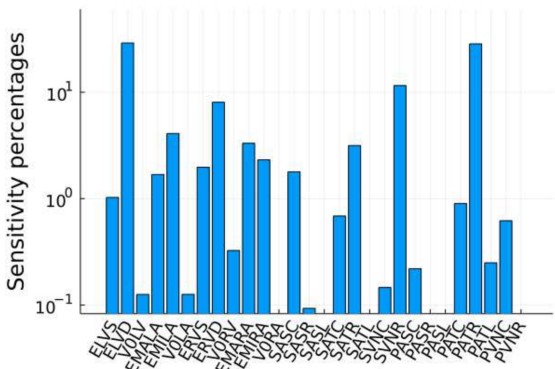

**Figure 4.** Model parameter sensitivity percentages.

**Table 3.** Legend for sensitivity analysis results.

| | | | |
|---|---|---|---|
| ELVS | LV systolic elastance | SASL | SAS inertia |
| ELVD | LV diastolic elastance | SATC | SAT compliance |
| VOLV | LV unstressed volume | SATR | SAT resistance |
| EMALA | LA maximal elastance | SATL | SAT inertia |
| EMILA | LA minimal elastance | SVNC | SVN compliance |
| VOLA | LA unstressed volume | SVNR | SVN resistance |
| ERVS | RV systolic elastance | PASC | PAS compliance |
| ERVD | RV diastolic elastance | PASR | PAS resistance |
| VORV | RV unstressed volume | PASL | PAS inertia |
| EMARA | RA maximal elastance | PATC | PAT compliance |
| EMIRA | RA minimal elastance | PATR | PAT resistance |
| VORA | RA unstressed volume | PATL | PAT inertia |
| SASC | SAS compliance | PVNC | PVN compliance |
| SASR | SAS resistance | PVNR | PVN resistance |

From the results, it can be seen that the parameter with the largest effect on the mean PAP is the pulmonary arterial resistance (28.8%), followed by the left ventricular diastolic elastance (28.1%), systemic venous resistance (12%), and right ventricular diastolic elastance (8.3%). In total, 11 parameters were selected to be optimized in subsequent sections, along with the aforementioned initial conditions (Section 2.2). Equation (29) shows the important parameter vector:

$$\bar{\theta} = [\ R_{PAT}\ E_{LV,d}\ R_{SVN}\ E_{RV,d}\ E_{LA,min}\ E_{RA,max}\ R_{SAT}\ E_{RA,min}\ E_{RV,s}$$
$$E_{LA,max}\ E_{LV,s}\ P_{PVN}^{init}\ P_{SVN}^{init}\ P_{AS}^{init}\ P_{PAT}^{init}\ ] \tag{29}$$

The local sensitivity analysis was also performed for both diseased heart valve conditions, and the analysis identified the same parameters as shown in Equation (28), but with differences in the sensitivity percentages allocated to each parameter. Similarly, the local sensitivity analysis was performed using the systolic PAP as the objective. The same important parameters were identified, except for a single parameter pair. The left atrium's maximal elastance was not on the list of the most important parameters—the pulmonary arterial compliance. These parameters only contribute to approximately 1.5% of the variance in their respective pulmonary arterial pressure values. Therefore, it was decided to use the parameters identified with the mean PAP, since mean PAP is typically used to identify pulmonary arterial hypertension.

### 3.2. Healthy Cardiovascular System Results

Using the nominal model parameters shown in Tables 1 and 2, synthetic datasets (D1 and D2) were generated and used as pseudo-measurements. The proposed inference model was tasked to recover the important model parameters (Equation (28)), while the remaining parameters were fixed to their respective nominal values. The goal of this investigation was to quantify the effects of the addition of heart chamber volume data and optimizer selection on parameter inference and pulmonary waveform prediction accuracy.

To estimate the parameter inference errors, the absolute percentage error (APE) metric was used. The equation used to calculate the APE for the $i$th important parameter is shown in Equation (30):

$$APE = 100\% \cdot \frac{|\theta_i - \theta_{i,true}|}{\theta_{i,true}} \tag{30}$$

Table 4 contains the APEs calculated for the different important parameters using the different datasets and optimizers. Additionally, the mean APE (MAPE) is also provided. The results indicate that the addition of the heart chamber volume data (D2) significantly improves the inference accuracy for all of the applied optimizers. Studying Equation (2) and Figure 2, it becomes clear that the heart chamber volume trends indirectly contain information about the atrial upstream flow rates ($Q_{svn}$ and $Q_{pvn}$), which is needed to integrate the mass continuity equation to find the time-dependent changes in chamber volume. The indirect addition of these flow rates positively impacts the ability of the inference model to accurately predict the unknown model parameters.

**Table 4.** Absolute percentage errors (APEs—Equation (30)) per parameter for datasets generated using nominal model parameters.

| Parameters | SP | D1 + ADAM | D1 + CGD | D1 + L-BFGS (hybrid) | D2 + ADAM | D2 + CGD | D2 + L-BFGS (hybrid) |
|---|---|---|---|---|---|---|---|
| $E_{LV,s}$ | 1.42 | 7.1 | 3.2 | 8.8 | 0.4 | 1.7 | 2.1 |
| $E_{LV,d}$ | 28.18 | 10.3 | 9.9 | 3.4 | 2.0 | 2.0 | 1.3 |
| $E_{LA,max}$ | 1.62 | 155.7 | 141.7 | 109.6 | 5.3 | 4.3 | 1.3 |
| $E_{LA,min}$ | 4.11 | 89.3 | 86.3 | 64.1 | 5.8 | 3.8 | 1.9 |
| $E_{RV,s}$ | 2.04 | 1.9 | 1.4 | 0.2 | 1.0 | 2.0 | 0.8 |
| $E_{RV,d}$ | 8.33 | 12.8 | 29.9 | 26.7 | 1.9 | 2.6 | 2.1 |
| $E_{RA,max}$ | 3.43 | 22.1 | 46.5 | 41.3 | 1.8 | 2.4 | 0.9 |
| $E_{RA,min}$ | 2.39 | 12.4 | 14.0 | 13.9 | 1.7 | 1.4 | 2.3 |
| $R_{SAT}$ | 3.17 | 8.5 | 6.7 | 7.1 | 4.3 | 2.1 | 0.3 |
| $R_{SVN}$ | 11.95 | 20.6 | 10.3 | 18.6 | 14.3 | 14.7 | 12.6 |
| $R_{PAT}$ | 28.83 | 5.0 | 39.0 | 2.9 | 2.7 | 3.8 | 3.0 |
| $P_{PVN}^{init}$ | - | 3.8 | 23.8 | 17.4 | 4.4 | 6.3 | 8.0 |
| $P_{SVN}^{init}$ | - | 14.1 | 2.5 | 12.4 | 2.1 | 8.2 | 0.9 |
| $P_{PS}^{init}$, $P_{PAT}^{init}$ | - | 5.4 | 16.6 | 15.3 | 0.0 | 1.2 | 1.4 |
| MAPE | - | 26.4 | 30.8 | 24.4 | 3.4 | 4.0 | 2.8 |

Of the three model configurations trained using D2, the Adam-L-BFGS hybrid optimization approach resulted in the lowest overall MAPE, followed by the Adam optimization approach. For the hybrid optimization approach, all parameters had APEs between

0 and 5%, except for the systemic venous resistance and the initial pulmonary venous pressure parameters. Although these two parameter inference errors are high, they have a small effect on the ability of the model to accurately recover important PAP values, as shown in Table 5. It is interesting to note that the Adam optimization approach more accurately predicts the pulmonary pressures but has a higher overall MAPE compared to the hybrid optimization approach. An explanation of this is that the estimated pulmonary arterial resistance parameter has a lower APE for the Adam approach when compared to the value predicted using the hybrid optimizer approach. Nonetheless, since the hybrid optimizer produces the most accurate parameter estimates, it was selected for further studies involving diseased mitral and aortic heart valves.

**Table 5.** Pulmonary pressure predictions for nominal model parameters.

| PAP (mmHg) | True Values | D2 + ADAM | D2 + L-BFGS (Hybrid) |
| --- | --- | --- | --- |
| Diastole | 13.93 | 13.96 | 13.94 |
| Systole | 25.11 | 25.05 | 24.96 |
| Mean | 17.65 | 17.58 | 17.42 |

As shown in the results in Table 4, the addition of the heart chamber volume waveforms (D2) during the optimization phase of the model substantially lowered the achieved MAPE values. At first glance, one can assume that the transvalvular flow rate waveforms do not significantly contribute to successfully finding the unknown parameters. To quantify the effects of the valvular flow rates on the inference accuracy, an additional simulation was performed, using a dataset that contained only the chamber volume and systemic arterial pressure waveforms (called D3), but not the transvalvular flow rate waveforms. The results are shown in Table 6. The optimizer applied to generate the results in Table 6 was the hybrid ADAM + L-BFGS optimizer. The results indicate that achieved MAPE is slightly higher when compared to the D2 + hybrid model results (Table 4), demonstrating that the valvular flow rates are beneficial to the search algorithm. Furthermore, it is important to note that the predicted $R_{PAT}$ values are substantially higher in Table 6 when compared to the values of the D2 + hybrid model results in Table 4, again highlighting the importance of including the valvular flow rates during the optimization phase, since this resistance value directly influences the simulated arterial pressure waveforms, as seen in Equation (13). To show this, the diastole, systole, and mean PAP values—which were 13.9, 24.7, and 17.2 mmHg, respectively—were compared to the true and D2-acquired values in Table 5. These, results show that the omission of the valvular flow data does slightly affect the capability of the model to recover the important PAP values. If the valvular flow data should be omitted, they can only be determined through the application of the current approach to clinical data, which could be the focus of future work.

To highlight the robustness of the proposed hybrid optimization approach, the standard deviations used for noise generation were increased to the values shown below, and the above parameters were reoptimized using D2 and the hybrid optimizer.

$$\sigma_{V_{lv}} = 8 \text{ mL}, \ \sigma_{V_{la}} = 6 \text{ mL}, \ \sigma_{V_{rv}} = 7 \text{ mL}, \ \sigma_{V_{ra}} = 5 \text{ mL}, \ \sigma_{P_{SAT}} = 10 \text{ mmHg}$$

$$\sigma_{Q_{AO}} = 61 \text{ mL/s}, \ \sigma_{Q_{PO}} = 37 \text{ mL/s}, \ \sigma_{Q_{MI}} = 50 \text{ mL/s}, \ \sigma_{Q_{TI}} = 39 \text{ mL/s}$$

The achieved MAPE when using the hybrid optimizer along with D2, with increased noise, was 4.7%. This MAPE value is similar in magnitude to the entry in Table 4 for the same optimizer and dataset used. For the remainder of this work, the datasets using the lower noise standard deviations are applied.

**Table 6.** Absolute percentage errors (APEs—Equation (30)) per parameter using a dataset containing only ventricle volume and systemic arterial pressure waveforms (D3).

| Parameters | APE |
|:---:|:---:|
| $E_{LV,s}$ | 2.0 |
| $E_{LV,d}$ | 2.0 |
| $E_{LA,max}$ | 2.4 |
| $E_{LA,min}$ | 2.0 |
| $E_{RV,s}$ | 2.2 |
| $E_{RV,d}$ | 4.0 |
| $E_{RA,max}$ | 3.6 |
| $E_{RA,min}$ | 2.1 |
| $R_{SAT}$ | 3.8 |
| $R_{SVN}$ | 13.3 |
| $R_{PAT}$ | 12.0 |
| $P_{PVN}^{init}$ | 1.2 |
| $P_{SVN}^{init}$ | 0.9 |
| $P_{PS}^{init}$ , $P_{PAT}^{init}$ | 1.7 |
| MAPE | 3.8 |

### 3.3. Diseased Heart Valve Case Studies

To investigate the ability of the proposed model to infer model parameters and PAP values for diseased heart valve cases, additional datasets were generated. These datasets consisted of data generated for induced aortic stenosis, mitral regurgitation, and a combination of these two valve diseases. To simulate aortic stenosis, the maximal valve opening angle was limited to 49.4°, which corresponds to a valvular flow area of 1 cm² for a valve diameter of 24.7 mm. Mitral regurgitation is induced by limiting the minimal mitral valve's closing angle to 33°, which corresponds to an open flow area fraction of 5%. For the combined case, both the aortic stenosis and mitral regurgitation limits were induced simultaneously.

For the three valvular disease cases, the nominal parameters listed in Tables 1 and 2 were used to generate the synthetic data. The APEs calculated for the five parameters with the highest *SP* values can be found in Table 7, along with MAPEs calculated using all of the parameters and initial conditions. The MAPE results show that the selected approach using D2, along with the hybrid optimizer, can predict the unknown model parameters for the three diseased cases with the same relative accuracy compared to the MAPE value calculated for the healthy case (Table 4). The case with the highest MAPE and highest parameter APE ($R_{SVN}$ = 12.12%) is the mitral regurgitation case, whereas the results show that the inclusion of aortic stenosis decreases the predicted parameter errors.

**Table 7.** Absolute percentage errors (APEs—Equation (30)) per parameter for datasets generated using nominal model parameters with aortic stenosis, mitral regurgitation, and both valvular diseases present.

| Parameters | Aortic Stenosis | Mitral Regurgitation | Combined |
|:---:|:---:|:---:|:---:|
| $E_{LV,d}$ | 1.26 | 0.62 | 1.93 |
| $E_{LA,min}$ | 1.44 | 0.74 | 1.82 |
| $E_{RV,d}$ | 0.05 | 1.87 | 2.03 |
| $R_{SVN}$ | 0.39 | 12.12 | 7.14 |
| $R_{PAT}$ | 2.80 | 7.74 | 6.62 |
| MAPE (all parameters) | 1.70 | 3.78 | 3.33 |

Figure 5 shows the right ventricle and pulmonary arterial pressure waveforms simulated using the predicted model parameters for the different diseased heart valve cases. The systole, diastole, and mean PAP predicted, along with the true values, are shown in Table 8, where the true values are the pressure values with no noise present. The results show that using the inferred model parameters, the 0D cardiovascular model can capture the true waveforms generated with the nominal parameter set with relative accuracy for both the unobserved ventricle and pulmonary artery pressures. For the combined diseased

case, it can be seen that the model using the inferred parameters slightly underpredicts the average pulmonary pressure prediction (average calculated using mean, systole, and diastole values) by approximately 1.5%, whereas for the other cases the model can accurately capture the diastole, systole, and mean PAP values.

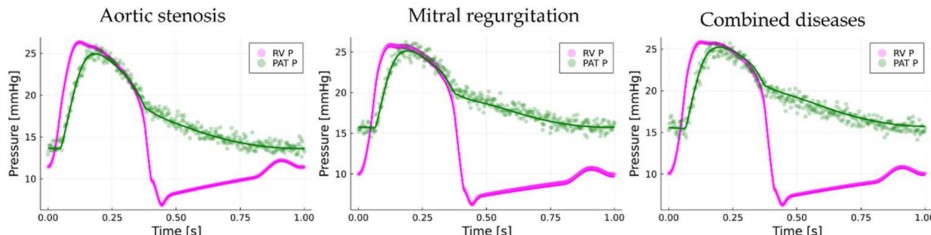

**Figure 5.** Pulmonary arterial and right ventricle pressure waveforms predicted by the model, along with predicted important PAP values using nominal model parameters. Solid lines—predicted values; markers—actual waveforms + noise.

**Table 8.** Global results for diseased heart valve cases (values in parentheses are the true pseudo-measurement values).

| Pulmonary Arterial Pressure | Aortic Stenosis | Mitral Regurgitation | Combined |
|---|---|---|---|
| Diastole | 13.61 (13.6) | 15.68 (15.67) | 15.48 (15.47) |
| Systole | 24.95 (25.03) | 25.12 (25.2) | 25.24 (25.12) |
| Mean | 17.33 (17.44) | 18.85 (19) | 19.11 (18.86) |

As seen in the results in Table 8, the mean PAP values for the different cases are below 25 mmHg [5], which is the typical upper limit for normal pulmonary pressures. To induce pulmonary hypertension, the pulmonary resistance was increased from 0.05 to 0.25 $\left[\frac{\text{mmHg·s}}{\text{mL}}\right]$, and the pseudo-data (e.g., transvalvular flow rates, heart chamber volumes, and systemic arterial pressures) were regenerated for the mentioned diseased heart valve settings. Using these new datasets, the inference model was again applied to recover the unknown model parameters, the goal being to investigate the model accuracy for hypertensive conditions. Table 9 contains the top five most important parameter APEs along with the MAPE using all of the predicted parameters. The results again indicated that the inference model can find the unknown important parameter values with relative accuracy, and that the effect of increased pulmonary arterial resistance on the overall MAPE values is not substantial. It is interesting to note that the pulmonary arterial resistance APEs from Table 9 are on average approximately 3.5% higher when compared to the values in Table 7, where the nominal pulmonary resistance value is used for pseudo-measurement generation.

**Table 9.** Absolute percentage errors (APEs—Equation (30)) per parameter for datasets generated using increased pulmonary arterial resistance with aortic stenosis, mitral regurgitation, and both valvular diseases present.

| Parameters | Aortic Stenosis | Mitral Regurgitation | Combined |
|---|---|---|---|
| $E_{LV,d}$ | 1.63 | 0.76 | 0.58 |
| $E_{LA,min}$ | 1.80 | 1.23 | 0.79 |
| $E_{RV,d}$ | 4.29 | 0.52 | 0.80 |
| $R_{SVN}$ | 0.07 | 2.58 | 5.06 |
| $R_{PAT}$ | 6.72 | 4.95 | 10.39 |
| MAPE (all parameters) | 3.13 | 2.51 | 3.42 |

Figure 6 shows the pulmonary arterial and right ventricular pressure waveforms along with the predicted and actual PAP values. These results show that although the predicted APEs for $R_{PAT}$ are higher for the increased resistance cases, the inference model is still capable of capturing the pulmonary pressure dynamics with relative accuracy.

However, for the combined case, it should be noted that the model slightly underpredicts the systolic right ventricular pressure, due to the overprediction of the right ventricular systolic elastance parameter (1.21 mmHg/mL vs. 1.15 mmHg/mL), which lowers pressure generation in the ventricle.

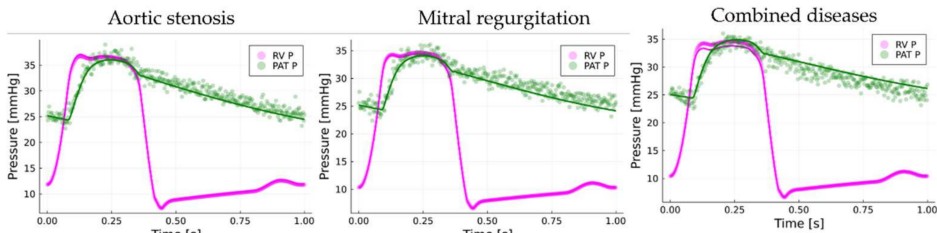

**Figure 6.** Pulmonary arterial and right ventricular pressure waveforms predicted by the model, along with predicted important PAP values using increased pulmonary arterial resistance. Solid lines—predicted values; markers—actual waveforms + noise.

For the PAP systole, diastole, and mean values shown in Table 10, the inference model has average error percentages of 1.12%, 2.49%, and 2.14%, respectively. These low errors highlight the possible ability of the proposed model to capture pulmonary pressures for diseased heart valves and hypertensive conditions.

**Table 10.** Global results for diseased heart valve cases with induced PAH (values in parentheses are the true pseudo-measurement values).

| Pulmonary Arterial Pressure | Aortic Stenosis | Mitral Regurgitation | Combined |
|---|---|---|---|
| Diastole | 24.33 (24.5) | 24.15 (24.5) | 24.6 (24.3) |
| Systole | 36.04 (34.4) | 34.08 (34.48) | 34.97 (34.44) |
| Mean | 29.57 (30.02) | 28.5 (29.15) | 29.85 (29.07) |

## 4. Conclusions

In the present work, it was demonstrated that the proposed algorithm can successfully recover the pulmonary arterial pressure waveform and associated clinically important values (i.e., systolic, diastolic, and mean values) using non-invasive measurements and a 0D cardiovascular dynamic network model. It was demonstrated that using systemic arterial pressure and heart chamber volume waveforms the proposed model can successfully recover simulated pulmonary arterial pressure waveforms. Furthermore, using the mentioned data in conjunction with an Adam-L-BFGS optimizer and the 0D cardiovascular network model yielded the most accurate results compared to other optimizers, such as conjugate gradient descent. It was found that including the valvular flow rates only slightly increased the inference accuracy of the proposed model. It should be noted that a limitation of the proposed approach is that it is assumes that the 0D model is complex enough to capture the dynamics of an actual human cardiovascular system—not only for synthetic data generation, but also for inference purposes. Therefore, future work will entail using retrospective clinical data to validate the proposed inference modelling approach.

**Author Contributions:** Conceptualization: R.L., P.G.H. and J.L.; methodology: R.L.; software: R.L.; original draft preparation: R.L., J.V.D.M., P.G.H. and J.L.; writing—review and editing: R.L., J.V.D.M., P.G.H. and J.L.; formal analysis: R.L. All authors have read and agreed to the published version of the manuscript.

**Funding:** This research received no external funding.

**Conflicts of Interest:** The authors declare no conflict of interest.

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
