# Peer review of "Estimation of Pulmonary Arterial Pressure Using Simulated Non-Invasive Measurements and Gradient-Based Optimization Techniques"

_mca, doi:10.3390/mca27050083_

Round 1

Author Response

The authors would like to extend their appreciation to the reviewer for her/his willingness to review our work and for the insightful questions asked. Please see below our responses to your queries.

Query 1:

“Introduction – paragraph 1: By the author’s assertion, pulmonary hypertension is a predictor of rheumatic heart disease. So, is quantifying pulmonary arterial pressure acting as indicator of PH? If so, what level of PAP indicates that your model has achieved PH? It seems to me that the authors are simulating specifically pulmonary arterial hypertension, so is PH in a general a predictor of RHD or is there a particular type of PH (namely, PAH) that predicts RHD?

Response:

As mentioned in paragraph 1, PH is specifically an independent predictor of mortality in RHD cases, in other words, if a patient has RHD and PH the likelihood of mortality is higher compared to patients with no signs of PH. Therefore, PH is important to know before clinicians are to take corrective actions.

In the present work PH was artificially induced in the model and the algorithm used to estimate the PAP for healthy and diseased cases to gauge the model setup accuracy. Therefore, the goal of the current work is to simply to recover the pulmonary arterial pressure waveforms using simulated non-invasive data, for healthy and diseased heart valve cases.

To clarify the reviewer’s query the following changes were made to paragraph 1:

“In rheumatic heart disease, the presence of PAH is an independent predictor of mortality [3], therefore, patients diagnosed with RHD and that have PH are at higher risk compared to patients with RHD and no PAH.

Additionally, all instances of PH are changed to PAH.

Query 2:

“Introduction – paragraph 4: The authors assert that “finite differences leads to computationally expensive and numerically unstable results”. It is surprising to this reviewer that the approximation of the gradient using finite differences would yield drastically different results and instability than using automatic differentiation. In my personal experience, instability typically results from using step sizes that are too small, which should be balanced with the time steps taken in the data (or pseudo data in this case).  Can authors clarify how and/or why using a finite difference approximation of the gradient specifically would be disadvantageous?

Response:

In previous work, the authors showed that using finite-differences when performing parameter estimation of a cardiovascular system model leads to longer run times and noticeably higher errors when compared to autodiff. If the reviewer is interested, they can see the following preprint, [1], for more results and information. The main reason for this observation according to the authors is the error accumulation that occurs when the gradients are calculated through the individual time step iterations of the ODE solver when the finite differencing steps are too large. For each variable time step taken by the solver (which in the present work is in the order of 2E5 time steps) the numerical error grows, which causes an increase in numerical noise in the results. Alternatively, if the step size is too small, it can lead to precision inaccuracies in the numerics of the program. Furthermore, inaccuracies in the ODE solution can be amplified by the numerical approximation of the FDM. This was also pointed out by [2]. The following was added to the paper to clarify:

“Numerical instability is likely due to the amplification of ODE solution errors and noise through the finite difference approximations. Furthermore, for variable time step ODE solvers, such as the one used in the present work, finite differences can lead to incorrect derivative estimates due to the different number of time steps used in the perturbed value evaluation, , and actual value evaluation, .

A very good online article to read on this topic:

https://uk.mathworks.com/help/optim/ug/optimizing-a-simulation-or-ordinary-differential-equation.html#btfb7f8

Query 3:

“Figures 1-4 have insufficient captions, and Figures 5 and 6 could be elaborated more. Personally, this reviewer thinks that figures should be able to stand mostly on their own with captions that clarify any undefined symbols, traces, curves, etc.

Response:

Proposed changes to figures 1-6 have been included as far as possible. Please see the revised article for the changes. Specific changes are discussed below.

Comment C: Figure 3, was changed to show the simulation outputs for the signals mentioned by the reviewer. Note the baseline results were not shown, seeing as the provided plots (figure 3) are simply baseline signals plus the added artificial noise. The following has been added to the text to clarify.

“Figure 3 below shows the data generated for the four non-hypertensive cases analyzed in the present work, namely the healthy case, aortic stenosis (AS) case, mitral regurgitation (MR) case and combined AR and MR case. It should be noted, only the ventricle volume changes (LV vol, RV vol, LA vol and RA vol), valvular flow rates (AO Q, PO Q, MI Q and TI Q) and systemic arterial pressure (SAT P) is used as pseudo-measurements during pulmonary pressure inference, and the ventricle pressures and pulmonary pressures are merely shown for the sake of completeness.”

Comment C: “Furthermore, I am having difficulty determining what the authors have done to create D1 and D2 (more on that below). Therefore, I think that (potentially) multiple subfigures could clarify what the pseudo data for each D1 and D2 is.”

Dataset 2 (D2) simply contains the ventricle volume changes along with the valvular flow rates and systemic arterial pressure, where D1 does not contain the volume changes. The following was changed in the text to clarify.

“The second dataset, D2, in addition to the transvalvular flow rates and systemic arterial pressure, contains the heart chamber volume changes”

Comment F: As requested by the reviewer all the baseline graphs have been added to figure 3. The authors believe the current layout is adequate seeing as figure 5 shows the recovered pulmonary and RV pressure waveforms for the three analysed cases (AS, MR and combined AS and MR), and figure 6 shows the same pressure waveforms but with induced PH.

Query 4:

“The manuscript refers to the model as a DAE; however, I see no equations that would result in a DAE from those provided. From what has been presented, this is a system of ODEs. If a DAE arises, please incorporate the equation(s) in the manuscript.

Response:

Corrected.

Query 5:

“Section 2.2 paragraph 2: I agree that the use of pseudo data generated from the model can elucidate the identifiability of the parameters. However, what is the purpose of these 2 different data sets? I find this very confusing, even how the two data sets are defined.

Response:

The reviewer is correct D2 contains the valvular flow rates, systemic arterial pressure and the chamber volume waveforms, whereas D1 only contains the valvular flow rates and systemic arterial pressure waveforms. As mentioned in Query 3 C – text was added to section 2.2 to clarify this confusion.

To highlight the effect of the valvular flow rates on the inference accuracy an additional simulation was performed with a dataset which contained the systemic arterial pressure and chamber volume waveforms only. The following was added to the paper:

“As seen in the results in Table 4, the addition of the heart chamber volume waveforms (D2) during the optimization phase of the model, substantially lowers the achieved MAPE values. At first glance, one can assume that the transvalvular flow rate waveforms do not significantly contribute to finding the unknown parameters. To quantify the effect of the valvular flow rates on the inference accuracy an additional simulation was performed, using a dataset that only contains the chamber volume and systemic arterial pressure waveforms and not the transvalvular flow rate waveforms. The results are shown below in Table 6. The results indicate that the achieved MAPE is higher when compared to the D2 + hybrid model results (Table 4), thus demonstrating that the valvular flow rates are beneficial to the search algorithm. Furthermore, it is important to note that the predicted R_SVN and R_PAT are substantially higher when compared to the values of the D2 + hybrid model results in Table 4, again highlighting the importance of including the valvular flow rates during the optimization phase.”

Query 6:

“Section 3.3: What more pseudo data sets are created? Are they the same as D1 and D2 for the disease states? If so, why both? Especially since the previous section showed that D2 contained the necessary information to inform mean PAP. How many data sets were created for each disease state? ”

Response:

The additional dataset is similar to D2 but simply generated with a higher pulmonary arterial resistance parameter as mentioned in section 3.2.

“To induce pulmonary hypertension the pulmonary resistance was increased from 0.05 to 0.25  and the pseudo data regenerated (transvalvular flow rates, heart chamber volumes and systemic arterial pressures) for the mentioned diseased heart valve settings.”

Query 6:

This is a stylistic preference and not as imperative to address. However, when reading this manuscript, it was very cumbersome to define parts of the equations and cite the equation before ever showing the equation. To this reviewer, it seems as though it should be the opposite. I spent some time looking beyond the current paragraph for an equation then having to go back to find my place, rather than reading linearly.

Response:

Thank you for the input, but the authors believe the current layout is adequate. No changes made.

Query 7:

“Tables 1 and 2 should have the parameter symbols and definitions included.”

Response:

Added as requested by reviewer.

Query 8:

“Is Equation 18 missing an initialization for beta_v?”

Response:

Corrected.

Query 8:

“Lines 299-302: There are several hyperparameters assigned, including epsilon, beta_1, beta_2, and eta. Can the authors elaborate on why these values were assigned? ”

Response:

These are the default values typically used with the Adam optimizer within the Flux.jl package. A reference was added to the paper.

Query 9:

“Line 330: Are the 11 parameters chosen for optimization the ones that explain 95% of the variance of the mean PAP as described in the Methods (Line 264)? ”

Response:

The reviewer is correct. The following has been changed in section 2.3:

“The top parameters that make up 95% of the variance in mean PAP are then selected as the important parameters to be optimized.”

Query 10:

Table 3: Why are P_PS_init and P_PAT_init together? Are they the same value?

Response:

Yes, for initialization purposes it was assumed that ,  are equal. The following was added to section 2.2.

Seeing as there is no substantial pressure drop between the pulmonary sinus and the pulmonary artery, the initial pulmonary arterial pressure and pulmonary sinus pressures were assumed to be equal, .”

Query 11:

“Line 412: Are the units for the resistance, correct?”

Response:

Corrected.

References

[1]         R. Laubscher, J. van der Merwe, J. Liebenberg, and P. Herbst, “Non-invasive estimation of left ventricle elastance using a multi-compartment lumped parameter model and gradient-based optimization with forward-mode automatic differentiation,” May 2022, [Online]. Available: http://arxiv.org/abs/2205.12330

[2]         A. L. Marsden, “Optimization in cardiovascular modeling,” Annu Rev Fluid Mech, vol. 46, pp. 519–546, Jan. 2014, doi: 10.1146/annurev-fluid-010313-141341.

Author Response

The authors would like to extend their appreciation to the reviewer for her/his willingness to review our work and for the insightful questions asked. Please see below our responses to your queries.

Query 1:

“The authors have titled the manuscript to suggest machine learning algorithms are used here. While “machine learning” is certainly a blanket term most often, I anticipated seeing a use of purely statistical models (e.g., Gaussian processes, neural networks, nonlinear regression) which is not present. I would suggest changing “scientific machine learning techniques,” to “statistical inference techniques” to avoid confusion.”

Response:

The title has been changed upon request from the reviewer. The new title reads:

“Estimation of pulmonary arterial pressure using simulated non-invasive measurements and gradient-based optimization techniques”

Query 2:

“Table 1, page 5: The authors may want to consider using and for both the atria and ventricles to avoid confusion. Also, that authors should ensure in their bounds that when running any sensitivity analyses.

Response:

Reviewer 1’s proposal was implemented in table 1. Please see the manuscript for the changes, which include descriptive text in the table to eliminate any confusion.

Query 3:

“Page 5, lines 179-185: The authors should provide more detail on the use of the valve model here, especially considering the original work focused on aortic and mitral valve regurgitation. Additional details about parameterization and numerical solution to the problem are warranted.

Response:

The valve cusp dynamic model section was expanded to show the ODEs solved in the computer model. The following was changed in the paper:

Typically, heart valves are modelled as simple diodes in 0D cardiovascular system models. For diode models the valve opening and closure processes are assumed to be instantaneous, therefore, the inertia of the valve cusps (leaflets) are ignored. Ignoring the valve leaflet motion for diseased heart valves cases can lead to the prediction of higher right ventricle and pulmonary arterial pressures as shown by [1]. Therefore, to accurately capture the pressure drop and thus fluid flow through the valve, the leaflet motion should be included in the system model. The valve model implemented in the current work stems from the paper by Korakianatis and Shi [1] which includes the simulation of the valve leaflet motion by solving an angular momentum equation for each heart valve. The heart valve parameters used in the present work was tuned by the previously mentioned authors to replicate actual pressure and flow waveforms in an adult human.”

“To ensure the selected ODE integrator can solve the dynamics of the valve cusp, the second-order angular momentum equation for the valve dynamics is expressed as two ODEs as shown in Equation 10.”

See equation 10.

(10)

Details regarding the numerical solution are provided in section 2.1 below table 2. In the current work, the BS 3/2 Runge-Kutta solver was used, and the relative tolerance and absolute tolerance were set to 1E-4 and 1E-6 respectively.

Query 4:

Page 6, lines 203-209: It isn’t clear to me why this system of equations is considered a DAE. What algebraic constraint is being enforced in the system?

Response:

Oversight on the side of the authors. All instances of “DAE” has been changed to “ODE”.

Query 5:

“Page 7, lines 222-227: The authors should motivate the use of a 3% error in generating a noisy signal.”

Response:

The noise percentages were arbitrarily chosen by the authors. In hindsight a better demonstration of the model robustness should be included in the paper. Therefore, an additional simulation was performed with significantly larger amount of noise in the pseudo measurements, and the parameters reoptimized. The following was added to the end of section 3.2:

To showcase the robustness of the proposed hybrid optimization approach, the standard deviations used for noise generation was increased to the values shown below, and the above parameters reoptimized.

 (See section 3.2 for standard deviations.)

The achieved MAPE when using the hybrid optimizer along with D2 with increased noise is 4.7%. This MAPE value is similar in magnitude to the entry in Table 4 above for the same optimizer and dataset used. For the remainder of the work, the datasets using the lower noise standard deviations will be applied.

Query 6:

“Is there a reason the authors ONLY considered adding noise to data using a fixed set of parameters? Why not randomly perturb the parameters to generate new signals, add noise, and THEN infer the parameters? That would provide a more convincing result, since the results here are biased to the single sets of data-generating parameters.”

Response:

These parameters are nominal parameters for a typical human cardiovascular system taken from literature [2], [3]. A similar approach was followed by Borgnakke et al. [4] in a recent publication. Future work will entail applying the proposed approach on actual patient data, which would then highlight any inability of the model to generalize to a wider range of parameters. The focus of the present work is simply to demonstrate the model accuracy and investigate the effect of the model settings and data availability on the achieved accuracy.

Query 7:

“Page 8, section 2.3: The authors need to justify why the local sensitivity analysis is only conducted on mPAP. The model is being calibrated to valve data in these pseudo-experiments, hence you should determine which parameters are most influential on the outputs you will calibrate your model to.

Response:

The authors selected the mean PAP seeing as it is a good representation of both the systole and diastole PAP values. To prove this, the local sensitivity analysis was rerun, but with the systole PAP as the objective.

The following is added to the paper (section 3.1):

“Similarly, the local sensitivity analysis was performed using the systolic PAP as objective. The same important parameters were identified except for a single parameter pair. The left atrium maximal elastance was not in the most important parameters list, rather the pulmonary arterial compliance. These parameters only contribute to approximately 1.5% of the variance in the respective pulmonary arterial pressure value. Therefore, it was decided to use the parameters identified with the mean PAP, seeing as mean PAP is typically used to identify pulmonary arterial hypertension.

Query 8:

“Page 9, line 289-290: “To speed up… were normalized using min-max scaling.” The authors need to be explicit here. If you are normalizing using the data, explicitly state how that is done for each measurement in the cost.”

Response:

Section 2.4 expanded as requested by reviewer.

The scaling transformation of the optimization parameters can be seen in Equation 24, where is the scaled parameter vector, is a vector of lower boundary parameter values and is a vector of upper boundary parameter values.

See equation 24.

(24)

The measurements and model predictions were scaled using the maximum and minimum measured values, e.g., for parameter i, max(x_tilde_i) and min(x_tilde_i). For example, Equation 25 shows the scaling of the systemic arterial pressure input waveform for time step.

See equation 25.

(25)

Query 9:

The authors should state these equations in a pseudo algorithm framework rather than as equations. That will make it easier to interpret what the arrows in eq. (24) and (25) represent.

Response:

Figure 1 in the manuscript serves to illustrate the framework of the algorithm in lieu of pseudocode. This is augmented by the equations which detail the underlying theory of the approach. The authors believe the current layout is adequate.

Query 10:

“Page 10, figure 4: Why don’t the authors consider the initial conditions or the valve parameters in the sensitivity analysis? The valve should play a role in calibrating the model to data, so it should be investigated and/or stated that this parameter wouldn’t change in the valve diseases considered.”

Response:

In the work of [2], the authors showed that by using fixed valve parameters, a range of valve diseases can be adequately modelled. Although these valve parameters would change depending on the valve disease type and its severity, the applied values are a good approximation to be used over a range of valve diseases, as shown in [5]. Future work would investigate the effect of incorporating a more fundamental valve model, which is not a function of a flow coefficient.

Query 11:

“Pages 11 table 3: It’s not clear what the values in the table represent. Why is a comma “,” used?”

Response:

Commas replaced with periods. Captions of tables updated.

Query 12:

“Page 11, lines 366-368: “It is interesting … but has a higher overall MAPE compared to the hybrid optimization.” This is likely due to identifiability issues (e.g., parameters are compensating to give a better fit). The authors should search the literature or at least state the possibility of non-identifiable parameters given the limited data and model structure.”

Response:

The authors do not believe this is the case, seeing as the data is generated using the cardiovascular model, therefore, all information generating mechanisms are present and exposed. Even with only selecting 11 parameters to be optimized one can argue that the remaining parameters are not observed. But in the present work the non-optimized parameters were held constant through-out the data generation and parameter optimization phases.

Query 13:

“Page 14: The conclusion is section 5, but the methods are section 3. Is there a discussion section missing in this manuscript?”

Response:

Corrected.

References:

[1]         T. Korakianitis and Y. Shi, “A concentrated parameter model for the human cardiovascular system including heart valve dynamics and atrioventricular interaction,” Med Eng Phys, vol. 28, no. 7, pp. 613–628, 2006, doi: 10.1016/j.medengphy.2005.10.004.

[2]         T. Korakianitis and Y. Shi, “Numerical simulation of cardiovascular dynamics with healthy and diseased heart valves,” J Biomech, vol. 39, no. 11, pp. 1964–1982, 2006, doi: 10.1016/j.jbiomech.2005.06.016.

[3]         T. Korakianitis and Y. Shi, “A concentrated parameter model for the human cardiovascular system including heart valve dynamics and atrioventricular interaction,” Med Eng Phys, vol. 28, no. 7, pp. 613–628, Sep. 2006, doi: 10.1016/j.medengphy.2005.10.004.

[4]         N. L. Bjørdalsbakke, J. T. Sturdy, D. R. Hose, and L. R. Hellevik, “Parameter estimation for closed-loop lumped parameter models of the systemic circulation using synthetic data,” Math Biosci, vol. 343, Jan. 2022, doi: 10.1016/j.mbs.2021.108731.

[5]         R. Laubscher, J. Liebenberg, and P. Herbst, “Dynamic simulation of aortic valve stenosis using a lumped parameter cardiovascular system model with flow regime dependent valve pressure loss characteristics,” Apr. 2022, [Online]. Available: http://arxiv.org/abs/2204.03701

Round 2

Author Response

Response to reviewers

Reviewer 1:

The authors would like to extend their appreciation to the reviewer for her/his willingness to review our work and for the insightful questions asked. Please see below our responses to your queries.

Query 1:

“Which optimizer was used for Table 6? For comparison, it would be nice to see all 3 optimizers reproduced for this new data set (maybe it could be called D3? I’m going to use this moving forward). In fact, it seems natural for Tables 4 and 6 to be merged showing all 3 optimizers with each of the 3 data sets.”

Response:

The hybrid (ADAM + L-BFGS) optimizer was applied to generate the results seen in table 6. The authors believe that table 6 should be separate from table 4, seeing as it only uses the best performing optimizer to investigate the effect of removing the valvular flow data not the effect of optimizer selection. The goal being to investigate if the inclusion of valvular flow data significantly affects the selected optimizer. The following has been added to section 3.2:

“…pressure waveforms (called D3) and not the transvalvular flow rate waveforms. The results are shown below in Table 6. The optimizer applied to generate the results in Table 6, is the hybrid ADAM + L-BFGS optimizer.

Query 2:

“Table 4 shows an order of magnitude improvement of MAPE from D1 to D2. However, Table 6 shows essentially the same MAPE values for D3 (3.8) as D2 (2.8-4) from Table 4. To this reviewer, it seems that including valve flows as a part of the data set does not contribute substantially to the reduction of MAPE since the MAPE values for D2 and D3 are of the same magnitude. Thus, the addition of more data (namely the valve flows) doesn’t decrease the error any further. The authors themselves state in new text that when they included greater parameter standard deviations, the new “MAPE value is similar in magnitude to the entry in Table 4” (line 788), which perhaps gives confidence to the identifiability of this methodology. Would not a similar logic be used to conclude that the valve flows are not necessary for the given application? The authors should address this.”

Response:

The reviewer is correct, in that the addition of the valvular flow data does not significantly affect the overall MAPEs, but as mentioned the accuracy of the pulmonary arterial and systemic vein resistances are noticeably lower, and both these parameters directly influence the simulated pulmonary arterial pressure. Therefore, the valvular flow data should still be included. The following has been added to clarify:

“…highlighting the importance of including the valvular flow rates during the optimization phase since this resistance value directly influence the simulated arterial pressure waveforms as seen in equations 13.”

Query 3:

“The authors say that “the predicted R_SVN and R_PAT are substantially higher in Table 6 when compared to …. Table 4”. However, the MAPE for R_SVN across the board in Table 4 seems to not be informed well (ranging from 11.95-20.6) and in Table 6 the value 13.3 falls within that range. To this reviewer, these values indicate that perhaps R_SVN is not an identifiable parameter given the loss function in equation 23 (the second one). The authors should perhaps consider removing the discussion of R_SVN as an example of the lesser performance in Table 6. Also, R_PAT does show a greater APE, but it does not seem to impact the overall MAPE greatly, suggesting perhaps that the value of R_PAT does not substantially impact the loss function either. Both cases seem to be results of fallacies in the local sensitivity analysis itself. In particular, the LSA was conducted with respect to the mPAP rather than the loss function. Just because R_PAT and R_SVN contribute greatly to the mPAP does not imply that the changing these values would have a significant impact on the reduction of the loss function with respect to the pseudodata, which is most likely the reason that larger errors in R_SVN and R_PAT do not incur larger MAPE. Would it not be more appropriate to conduct the local sensitivity analysis with respect to the loss function to determine which parameters can be adequately informed by the data given?”

Response:

The reviewer is correct, the discussion regarding the systemic vein resistance inference error was removed seeing as the inference accuracy does not change significantly for the different model setups. The pulmonary resistance on the other hand would influence the capability of the model to recover the PAP waveform. To highlight this the D3 model was re-simulated, and the important diastole, systole and mean PAP values achieved, extracted, and compared to the values in Table 5. The following has been added to clarify:

“…highlighting the importance of including the valvular flow rates during the optimization phase since this resistance value directly influence the simulated arterial pressure waveforms as seen in equations 13. To show this the diastole, systole and mean PAP values which were 13.9, 24.7 and 17.2 mmHg respectively were compared to the true and D2 acquired values in Table 5. These, results show that the omission of the valvular flow data, does slightly affect the capability of the model to recover the important PAP values. If the valvular flow data should be omitted can only be determined through the application of the current approach to clinical data, which would be the focus of future work.”

The LSA is used to specifically identify parameters affecting the PAP waveforms, seeing as that are the desired values to be extracted from the sparse data using the model and the optimizer. Optimizing a parameter set informed by an LSA using the cost function would identify parameters which will only recreate the available waveforms, not necessarily accurately recover the PAP waveforms, since the available pressure reading is on the systemic branch of the cardiovascular system. Therefore, the parameter optimizer should be constrained to find only parameters affecting the waveform that should be recovered.

Query 4:

“The authors failed to address my previous concern about data needed to inform the model. To this reviewer, it seems like a great opportunity to discuss that the necessary data to inform PAP is the systemic arterial pressure and heart volumes. Is this the case? If so, can the authors comment further?”

Response:

The reviewer is correct, the systemic arterial pressure and heart chamber volume measurements are required as shown by the current work. The following has been added to the conclusions section:

“It was demonstrated that using systemic arterial pressure and heart chamber volume waveforms, the proposed model can successfully recover simulated pulmonary arterial pressure waveforms. Furthermore, using the mentioned data in conjunction with an Adam-L-BFGS optimizer with the 0D cardiovascular network model yielded the most accurate results compared to other optimizers such as conjugate-gradient descent. It was found that including the valvular flow rates only slightly increased the inference accuracy of the proposed model.”

Query 5:

“There are 2 equation 23’s.”

Response:

Corrected.

Query 6:

“Figure 3: Please make all similar axes the same size for better comparison between similar panels.”

Response:

Seeing as the images are quite small and they are separate cases, the current scale of the axis systems are adequate.

Reviewer 2 Report

The authors have satisfied a majority of my comments. The manuscript would benefit from an expanded limitations section that includes (1) the lack of testing on whether different parameters could be inferred by generating noisy pseudo-data (e.g., generating new data from random perturbations in parameters, and (2) the limitation that valve parameters, which should be patient specific depending on disease severity, are not including in the sensitivity analysis nor the inference procedures.

Otherwise, this manuscript is at a stage for acceptance and publication.

Author Response

Response to reviewers

Reviewer 2:

The authors would like to extend their appreciation to the reviewer for her/his willingness to review our work and for the insightful questions asked. Please see below our responses to your queries.

Query 1:

“…the lack of testing on whether different parameters could be inferred by generating noisy pseudo-data.”

Response:

The present work specifically focuses on capturing the PAP changes, and as shown the selected parameters are the dominant factors influencing this metric. Furthermore, the parameter values and ranges are selected to be representative of an actual human cardiovascular system, therefore, the parameters values are varied only within reasonable bounds and used to recover parameter combinations typically referenced in literature.

Query 2:

“…the limitation that valve parameters, which should be patient specific depending on disease severity…

Response:

The authors, agree that the presented valve modelling is lacking to a certain extent seeing as it is based on empirical values. But the solution would not be to vary the valve parameters but rather apply a more advanced valve modelling methodology as described in [1], this will be included in future work.

References:

[1]         R. Laubscher, J. Liebenberg, and P. Herbst, “Dynamic simulation of aortic valve stenosis using a lumped parameter cardiovascular system model with flow regime dependent valve pressure loss characteristics,” Apr. 2022, [Online]. Available: http://arxiv.org/abs/2204.03701

Round 3

Reviewer 1 Report

The authors have addressed my concerns, and I recommend this manuscript for acceptance for publication.